# Community Health Workers’ Knowledge, Attitudes, and Practices towards Epilepsy in Sofala, Central Mozambique

**DOI:** 10.3390/ijerph192215420

**Published:** 2022-11-21

**Authors:** Vasco Francisco Japissane Cumbe, Claire Greene, Afonso Mazine Tiago Fumo, Hélder Fumo, Dirceu Mabunda, Lídia Chaúque Gouveia, Maria A. Oquendo, Cristiane S. Duarte, Mohsin Sidat, Jair de Jesus Mari

**Affiliations:** 1Mental Health Department, Ministry of Health, Provincial Health Directorate of Sofala, Beira 543, Mozambique; 2Mental Health and Psychiatry Department, Faculty of Medicine, Eduardo Mondlane University (UEM), Maputo 257, Mozambique; 3Medicine Department, Psychiatry and Mental Health Service, Beira Central Hospital, Sofala 1613, Mozambique; 4Departamento de Psiquiatria, Escola Paulista de Medicina, Universidade Federal de São Paulo, UNIFESP, São Paulo 04017-030, Brazil; 5Program on Forced Migration and Health, Heilbrunn Department of Population and Family Health, Columbia University Mailman School of Public Health, New York, NY 10032, USA; 6Department and Institute of Psychiatry, Faculty of Medicine, University of São Paulo (USP), São Paulo 05403-903, Brazil; 7Service of Psychiatry and Mental Health, Mavalane General Hospital, Maputo 7981000, Mozambique; 8Department of Mental Health, Directorate of Public Health, Ministry of Health, Maputo 264, Mozambique; 9Department of Psychiatry, Perelman School of Medicine, University of Pennsylvania, Philadelphia, PA 19104, USA; 10Department of Psychiatry, College of Physicians and Surgeons, Columbia University, New York, NY 10032, USA; 11Department of Community Health, Faculty of Medicine, Eduardo Mondlane University (UEM), Maputo 257, Mozambique

**Keywords:** epilepsy, community health workers, knowledge, attitudes, practices, primary health care, Mozambique

## Abstract

Background: Epilepsy is the most common neurological disease in the world, affecting 50 million people, with the majority living in low- and middle-income countries (LMICs). A major focus of epilepsy treatment in LMICs has been task-sharing the identification and care for epilepsy by community health workers (CHWs). The present study aimed to assess the knowledge, attitudes, and practices (KAPs) of CHWs towards epilepsy in Mozambique. Methods: One hundred and thirty-five CHWs completed a questionnaire that included socio-demographic characteristics and 44-items divided into six subscales pertaining to KAPs towards epilepsy (QKAP-EPI) across nine districts of Sofala, Mozambique. The internal consistency was examined to evaluate the reliability of the instrument (QKAP-EPI). The association between sociodemographic variables and QKAP-EPI subscales was examined using linear regression models. Results: The internal consistency was moderate for two subscales (causes of epilepsy, α = 0.65; medical treatment, α = 0.694), acceptable for cultural treatment (α = 0.797) and excellent for 2 subscales (safety and risks, α = 0.926; negative attitudes, α = 0.904). Overall, CHWs demonstrated accurate epilepsy knowledge (medical treatment: mean = 1.63, SD = 0.28; safety/risks: mean = 1.62, SD = 0.59). However, CHWs reported inaccurate epilepsy knowledge of the causes, negative attitudes, as well as culturally specific treatments for epilepsy, such as: “if a person with epilepsy burns when set on fire they cannot be treated”. Knowledge about how to manage epileptic seizures varied across the different emergency care practices, from the accurate belief that it is not advisable to place objects in the individual’s mouth during an epileptic seizure, to the wrong perception of the need to hold the person in seizures to control seizures. Heterogeneity in the level of epilepsy knowledge was observed among CHWs, when considering epilepsy according to the local names as treatable (“Dzumba”) and other forms as untreatable (“Nzwiti”). Conclusion: CHWs knowledge of medical treatment and epilepsy safety/risks were adequate. However, information on the causes of epilepsy, stigmatizing attitudes, cultural treatment, and some knowledge of epileptic seizure management were low. These areas of poor knowledge should be the focus of educating CHWs in increasing their ability to provide quality care for patients with epilepsy in Mozambique.

## 1. Introduction

Epilepsy is the most common neurological disease in the world, affecting 50 million people, with the majority (85%) living in low and middle-income countries (LMICs) of whom 80% do not receive appropriate treatment [1,2,3]. The unavailability of treatment may be caused by lack of trained people and essential drugs or an inadequate care [3]. In Mozambique, neuropsychiatric disorders account for 23.1% of all Years Lived with Disability (YLD) in individuals aged 15–49 years [4], and epilepsy is the leading cause of outpatient mental health consultations, especially in rural areas [5,6,7]. The risk of premature death among people with epilepsy (PWE) is up to three times higher than for the general population [8]. Epilepsy is a highly stigmatized condition, particularly because it is often inaccurately considered contagious and stigmatized [1,8,9]. In low-resource settings, epilepsy is very often perceived as a mental illness or related to madness, resulting from spiritual possession or witchcraft, making PWE more likely to seek traditional/spiritual treatment instead of medical treatment [10]. If proper diagnosis and treatment were provided, up to 70% of PWE would live seizure free. Globally, between 50–75% of PWE are not treated, despite treatment costing less than USD 5 per year [11]. In Mozambique, the National Health Service offers care and treatment services free of charge and more than 90% of the population benefits from these services. Task shifting epilepsy’s diagnosis and management to community health workers (CHWs) can reduce the care and treatment access gap [12]. 

In 2014 the Mozambican Ministry of Health implemented a strategy of training CHWs to care for common health problems in regions with limited access to healthcare and placing at least one Psychiatric Technician at primary healthcare (PHC) facilities in all 135 districts nationally [6,13]. Supervised by Psychiatrists, Psychiatric Technicians are mid-level health professionals trained for promoting mental health prevention, diagnosis, and treatment of common mental health disorders, including epilepsy at the PHC level in Mozambique. When properly trained, CHWs work closely with Psychiatric Technicians in addressing community awareness about epilepsy, mental health problems, and establishing referrals from the community to PHC [13]. This expansion of mental health care nationwide allowed a greater integration of the mental health program into general health care and in raising awareness about the meaning of mental illness/epilepsy at the level of communities. This program also supported a greater connection with the community health workers in general, and supporting their role in connecting community patients to mental health care settings. There are multiple trials which have demonstrated that lay personnel can efficaciously treat common mental disorders using psychological interventions [14], however, there is a paucity of published studies describing CHWs role in the promotion of epilepsy treatment and care in LMICs. Few published studies in Mozambique or other Sub-Saharan African countries have explored CHWs knowledge, attitudes, and practices towards epilepsy. Most of the studies conducted in other African countries, related to epilepsy, have focused on health professionals, students, or communities in general. A study assessing Zambian health professionals’ knowledge regarding epilepsy showed a knowledge gap among nurses (25%) who often stated they would not allow their children to marry a person with epilepsy [15], a finding similar to that was reported in another study conducted in Nigeria among community residents [16]. Another qualitative study conducted in Congo has shown that community misperceptions about PWE are common [17]. In contrast, a study performed in Bolivia showed that the nonmedical professionals (nurses and community health workers) presented good baseline knowledge, attitudes, and beliefs related to epilepsy [18]. Like in many African countries, care delivered by Mozambican CHWs is often more culturally accepted and represents the first line of care in community settings [19]. They are also the first point of entry into PHC in Mozambique [20]. In Mozambique, however, it is unknown the extent to which CHWs have adequate knowledge, attitudes, and practices towards epilepsy to allow them to recognize and refer PWE to PHC and fulfill their new role of connecting community patients to epilepsy care in primary health care. This study aims to assess: (a) CHWs knowledge, attitudes, and practices towards epilepsy; (b) socio-demographic correlates of knowledge, attitudes, and practices towards epilepsy; and (c) the reliability and validity of the questionnaire of knowledge, attitudes, and practices towards epilepsy) (QKAP-EPI) among CHWs in Mozambique. This study represents the first phase of assessing the knowledge, attitudes, and practices of CHWs towards epilepsy before evaluating the effect of training CHWs using the WHO Mental Health Gap Action Program (MhGAP) epilepsy package, with the expectation of recommending MISAU for including this adapted package into national CHWs training curricula. The aim of this study is to assess the knowledge, attitudes, and practices (KAPs) of community health workers towards epilepsy in Mozambique.

## 2. Materials and Methods

### 2.1. Structure of Mental Health Care Systems in Mozambique

The National Mental Health Program is under the management of the Department of Mental Health at the Ministry of Health. The Provincial Mental Health Program organizes the districts data and reports to the National Mental Health Program in the Ministry of Health (MISAU) [21]. The country has 25 Psychiatrists (only 18 of whom are Mozambican), 305 Psychiatric Technicians, 130 Clinical Psychologists, and 14 Occupational Therapists who provide services to <10% of the public clinics (Mental Health Department, MISAU, 2019). Since the 1990s, Mozambique has been a leader in Sub-Saharan Africa in scaling-up the training of task-shared mental health professionals (Psychiatric Technicians) to provide treatment for all categories of mental illness and epilepsy [13,21,22].

### 2.2. Study Design and Setting

This cross-sectional study was conducted from January 2018 to July 2018 to assess CHWs knowledge, attitudes, and practices towards epilepsy in PHC in the province of Sofala. Sofala is a mostly rural province (59.1%) located in the central region of Mozambique with a population of 2,221,803 inhabitants and an illiteracy rate of 43.1%. There are 161 health facilities distributed across 13 districts of Sofala [23] providing PHC and epilepsy care. Mental health and epilepsy services are available in 25 health facilities with a coverage of 15% of the province [24]. Most mental health services are in district-level facilities, except Beira City which has mental health services in 11 out of its 20 health facilities [24].

### 2.3. Study Participants and Procedures 

Fifteen CHWs were selected purposefully from nine districts (total n = 135) in Sofala province (Figure 1). Districts were included if they had no previous implementation of World Health Organization Mental Health Gap Action Program (WHO mhGAP) and had no CHWs trained in the detection and management of epilepsy cases. In general terms, 15 CHWs are allocated to each district—if districts had more than 15 CHWs we randomly selected 15 from the larger group. The lead CHW in each health facility invited all CHWs at that health facility to participate in the study and there were no refusals reported. Participants were included if they were 18 years or older, belonged to the district headquarters, were fluent in Portuguese, and provided written informed consent. 

### 2.4. Study Instruments

The sociodemographic questionnaire administered to the participants included the variables such as gender, age, education, marital status, religion, occupation, and residence. The CHWs knowledge, attitudes, and practices towards epilepsy were evaluated through the QKAP-EPI (questionnaire of knowledge, attitudes, and practices towards epilepsy). The QKAP-EPI was translated from English to Portuguese, back translated and adapted to the context of Sofala Province from a Kenyan version [26]. The adaptation and creation of the QKAP-EPI, from the original English version, followed a structured process to ensure content, semantic, and technical equivalence. This focused initially on a series of translations of the English version with a focus on comprehensibility (does an item retain its original semantic equivalence), appropriateness (fit, compatibility with new cultural context), and a specific focus on ease-of-understanding given the low schooling achieved among CHWs in primary health care in Mozambique. The final instrument was field tested by administering it to 5 community health workers in two health facilities in Beira City, Mozambique. To the original scale of 5 subscales with 38 questions, we added an additional subscale of practices during seizures including 6 items. Thus, the questionnaire contains 44 items, divided into 6 subscales: *causes of epilepsy*, *medical treatment*, *cultural treatment*, *risks and safety*, *negative attitudes,* and *practices during an epileptic seizure*. Response options are in the form of a 3-point Likert scale, ranging from 0 (I do not believe), 1 (I believe very little), to 2 (I believe very much). The questionnaire was completed on average between 15–20 min. A higher overall score reflects a positive epilepsy knowledge, attitudes, and practices towards epilepsy. The higher score on items 1–14, 24–27, and 43 and 44 reflects positive epilepsy knowledge, attitudes, and practices. Lower score of the items 15–23 and 28–42 reflects positive epilepsy knowledge, attitudes, and practices. Total score ranges from 0 to 88 points.

### 2.5. Ethical Issues

Written consent was obtained from all eligible participants. The CHWs were assured their participation was voluntary and it would not affect their work. Approval for the study was obtained from the Eduardo Mondlane University (Medical School) & Maputo Central Hospital Ethical Review Committee under registration CIBS FM&HCM/74/2016.

### 2.6. Data Analysis

Descriptive analyses including the mean and the standard deviation (SD) for continuous demographic variables (e.g., age) and frequencies for categorical demographic variables (e.g., occupation, marital status, and schooling level achieved) were calculated to characterize the full sample of community health workers. We estimated the internal consistency of subscales using ordinal alpha coefficients. We evaluated the internal construct validity of the QKAP-EPI using a confirmatory factor analysis. Fit of the model was evaluated using the RMSEA (root mean square error of approximation), CFI (comparative fit index), TLI (Tucker Lewis’s index), SRMR (standardized root mean residual), and examination of the factor loadings and modification indices. The correlation between subscales was calculated using Pearson correlation coefficients. We used linear regression to examine demographic correlates of each subscale score including age, sex, marital status, schooling level achieved, occupation, religious affiliation, and district as covariates. Significance level was set at 0.05. All analyses were conducted in Stata 14.

## 3. Results

### 3.1. Sample Description

Of the 135 CHWs, most were male (69.3%), had a secondary or higher education level (55.6%), were Christian (70.4%), and either primarily unemployed (43.7%) or a farmer (23.0%). The average age was 38 years. (SD = 13.01; Table 1).

### 3.2. Internal Consistency of Knowledge, Attitudes, and Practices Regarding Epilepsy Subscales and Items with Low Item Rest Correlation

The internal consistency of the subscale practices during seizures was low (α = 0.238). The internal consistency was moderate for two subscales (causes of epilepsy, α= 0.65; medical treatment, α = 0.694), acceptable for cultural treatment (α = 0.797) and excellent for safety and risks (α = 0.926) and negative attitudes (α = 0.904). The overall ordinal alpha was not reported because the adapted questionnaire of knowledge, attitudes, and practices is a multidimensional scale. Several subscales contained item(s) that were not strongly correlated with other items in the subscale, suggesting that they may be less informative or specific to the subscale construct [27]; (Table 2). 

To examine the fit of the six-factor model of knowledge, attitudes, and practices, we conducted a confirmatory factor analysis. The fit of the original model was poor (RMSEA = 0.075, 90% CI: 0.069, 0.081; CFI = 0.582; TLI = 0.553; SRMR = 0.103), primarily due to the multidimensionality of the subscale practices during seizures. The first three items (*we should put a stick in the mouth of the PWE during seizures*; *we should give a drink to the PWE during seizures*; *we should stay away from PWE during seizures*) most strongly loaded on the first factor, now referred to as “*practices of sticking into a mouth, give a drink, stay away*”. The last three items (*we should hold PWE during seizures to stop seizures; we must help PWE during the seizures not to hit the head on the ground; we must place the PWE in the right lateral safety position to avoid aspiration*) most strongly loaded on the second factor, now referred to as “*practices of holding, helping, placing in safety position*”. Dividing the practices during seizures subscale into two factors and estimating the covariance of highly correlated items improved the model, displaying adequate fit (RMSEA = 0.059, 90% CI: 0.051, 0.066; CFI = 0.749; TLI = 0.727; SRMS = 0.098). The item, “*we should stay away from PWE during seizures”,* from the added “s*ticking into a mouth, give a drink, and staying away during seizures*” subscale loaded slightly more strongly on the negative attitude’s subscale. Although allowing cross-loading of this item onto both the “*sticking into a mouth, give a drink, and staying away during seizures*” and the negative attitudes subscale would have slightly improved the fit, the benefits were not substantial and may not offset the complexity introduced to scoring the QKAP-EPI and deviating from the original item domains.

### 3.3. CHWs Responses to the Adapted Questionnaire of Knowledge, Attitudes, and Practices towards Epilepsy

Most of the CHWs very much believed that causes of epilepsy may be related to head injury (59.3%), malaria/meningitis/fever (45.9%), and brain injury (54.8%), but some did not believe that it can be inherited (52.6%) or be related to childbirth injury (54.8%), denoting a lack of knowledge regarding these two items. They also did believe very much that: epilepsy can be treated (77.0%); treatment should occur on a regular basis to better control the disease (77.8%); all the conditions of epilepsy with local names (Nzwite, Njiri Njiri) are best treated by a doctor (75.6%; 71.8%); antiepileptic medicines control the seizures (71.9%); and these medicines can be available in health facilities (88.2%), demonstrating a positive knowledge regarding the medical treatment of epilepsy.

In the “*cultural treatment*” subscale, a proportion of CHWs endorsed false cures or treatments for epilepsy. For example, 51.1% of CHWs believe that if a PWE burns when set on fire, this means this person cannot be cured from epilepsy. Further, 21.0% believed that during a seizure, it would be appropriate to put water on the person or make the person smell shoes. Lastly, 23.0% believed that putting a stick in someone’s mouth during a seizure can treat epilepsy. More than half of CHWs believed that some forms of epilepsy with local names (Dzumba) can be treated while other forms (Nzwiti) cannot be treated (52.6%) demonstrating an inaccurate knowledge of how epilepsy should be treated according to the cultural perception of what the appropriate treatment of different types of epilepsy should be when using local terms. More than half of the CHWs believe that the legs of a PWE should be stretched during seizures (52.6%) which goes against the epilepsy treatment protocol. 

In the safety and risks subscale, more than half of the CHWs believed very much that PWE should not climb trees (74.04%), not drive (70.37%), avoid fire (85.19%), or water sites (79.26%), which demonstrates positive knowledge, attitudes, and practices towards epilepsy in safety and risks. More than half of the CHWs demonstrated low levels of negative attitudes and did believe that PWE should be able to: get married (65.9%), go to school (71.9%), or have a job (74.4%).

Most of the CHWs did not believe that one should put a stick in people’s mouths, give a drink, and stay away from PWE during seizures (73.3%, 78.5%, and 71.1%), demonstrating an accurate epilepsy knowledge. However, more than half of CHWs believed inaccurately that one should hold people during seizures to stop them (67.7%) which is against the recommended epilepsy treatment protocols. (Table 3).

### 3.4. Average of Subscales of the Adapted Questionnaire of Knowledges, Attitudes, and Practices towards Epilepsy Applied to 135 CHWs in Sofala, Mozambique

The average of subscales, which ranged from 0 (no knowledge) to 2 (good knowledge), were adequate for the following subscales: *medical treatment* (M = 1.63, SD = 0.28), *safety and risks* (M = 1.62, SD = 0.59), and *practices of holding*, *helping*, *and placing in safety position during seizures* (M = 1.60, SD = 0.48). The average of subscales *causes of epilepsy* (M = 1.10, SD = 0.53), *cultural treatment* (M = 1.48, SD = 0.39), *negative attitudes* (M = 1.47, SD = 0.48), and *practices of sticking into a mouth, give a drink, stay away during seizures* (M = 1.43, SD = 0.42) were slightly lower indicating inaccurate epilepsy knowledge (Table 4).

## 4. Discussion

The purpose of this study was to assess the CHWs knowledge, attitudes, and practices towards epilepsy across nine districts in Sofala province, Mozambique. This study was conducted prior to the implementation of a training in mhGAP to allow trainers to tailor their training activities. This training focused on building capacity for assessment and management of epilepsy within primary care and intended to be appropriate for and responsive to CHW’s current knowledge, attitudes, and practices. CHWs epilepsy knowledge of medical treatment and epilepsy safety/risks were adequate. However, knowledge related to the causes of epilepsy, negative attitudes, culturally specific treatment, and some aspects of epileptic seizures response were low. Knowledge regarding the causes of epilepsy was limited among CHWs relative to the other subscales. CHWs did not believe that epilepsy could be inherited or related to childhood injury, but correctly identified head injury, malaria/meningitis/fever, and brain injury as risk factors and this is against the false perception in an unpublished study conducted in the south of Mozambique, where most of the population reported believing that epilepsy was caused by “bad spirits” [28]. In our study CHWs displayed moderate to high levels of accurate epilepsy knowledge regarding medical intervention, epilepsy safety/risks, and some aspects of practices during epileptic seizures. Maintaining and strengthening these domains of knowledge regarding medical, safety/risks perceptions, and some aspects of practices during epileptic seizures can allow CHWs to facilitate linkages between the community and PHC. They can do this by highlighting the availability of conventional treatment of different epilepsy designations in the local health network and awareness of some dangerous practices during epileptic seizures. Cultural context is often complicated using different terms to designate epilepsy, generally associated with different causes [29] and may impact medical treatment negatively, when considering some of these different designations of epilepsy as treatable by traditional healers or in churches instead of going to the hospital. An unpublished study conducted in the south of Mozambique emphasized that the various designations of epilepsy such as “moon disease” and “spirit disease” may support beliefs and myths that traditional healers are the best individuals to treat epilepsy [30]. 

Some CHWs have adequate knowledge about some of the inaccuracies of the cultural treatments of epilepsy—evidenced by the non-acceptance of dangerous practices such as putting a stick in the mouth during seizures—a significant proportion of CHWs still believed that local names for epilepsy such as Dzumba or Nzwiti, could determine whether it could be treated or not. The differences of opinions related to treatments for epilepsy using local versus non-local terms demonstrates the risk of non-dissemination of the local terms that could be a barrier for CHWs performance in linking PWE in the community to PHC [29]. Other worrying perceptions were that many CHWs believed that individual’s knees should be stretched during a seizure or that people should be held during seizures, which are very dangerous practices for PWE and contrary to standardized epilepsy treatment protocols about proper conduct during epileptic seizures. Understanding the cultural beliefs gives insight into how people cope with epilepsy [31,32]. In the absence of this knowledge, misunderstanding between CHWs, PWE, and health professionals [33] may result in poor adherence to antiepileptics medicines. 

This study also revealed negative and stigmatizing attitudes held by a notable proportion of CHWs toward PWE. Many CHWs felt that PWE cannot or should not have a normal life and are a burden to their family. While most CHWs did not report these attitudes, other studies have found that PWE have often been marginalized in their societies with less opportunities for education, social relationships, employment, and marriage [29,34] as epilepsy is seen by the public as highly contagious and shameful [35]. In this study we found that there was some variability in knowledge about epilepsy by demographic factors and district, which may be explained in part by the different cultural contexts of each district. 

It is noteworthy that the current CHWs training package, in Mozambique, does not include epilepsy, which may be one obstacle to acquisition of positive knowledge, attitudes, and practices towards epilepsy, reinforcing the existing negative attitudes towards PWE. Given that most individuals trained as CHWs have strong cultural influence in the communities where they are inserted [18], they could be catalysts to improve “bridging” of PWE to PHC. The strength of CHWs being “integrated” into the community allow them to relate well to the local community but can be harmful if effective practices are not adequately instilled during training.

These findings must be interpreted considering several limitations. First, we used a Portuguese instrument that at some point may not have facilitated the comprehension of the instrument in its entirety by CHWs, because some of them may not fully understand Portuguese and may be more comfortable in their local language. Second, although adapted, this instrument has not been previously validated for the context of Mozambique. Last, the findings may not be generalizable to other areas in Mozambique due to contextual specificities of the studied area. These limitations may have eventually led to differences in interpretation of the knowledge, attitudes, and practices statements. For example, the “correct” response for some of the knowledge items, such as whether epilepsy is inherited and whether it can be caused by head injury have the potential to vary based on subtle differences in translation and interpretation. For these reasons, we suggest use of locally adapted and validated instruments in other regions of Mozambique in case of similar study intended. To account for this limitation, we evaluated the internal consistency and internal construct validity of this measure prior to evaluating the levels and correlates of knowledge, attitudes, and practices related to epilepsy. The low to moderate internal consistency may be due in part the subscales contained poorly correlated items, demonstrated by the low item rest correlation for items within the three sub-scales practices during crisis, causes of epilepsy, medical treatment, meaning that may be less informative or not specific to the construct in question. Third, this was an exploratory study with 135 community health workers. Given the small sample size and the descriptive, exploratory nature of the study, we did not correct for multiple comparisons, and thus we cannot make generalizations to the larger population of CHWs in Mozambique. 

This study also possessed several strengths. First, to our knowledge this is one of the few published studies to evaluate the knowledge, attitudes, and practices towards epilepsy among CHWs in Sub-Saharan Africa. Given the high prevalence and burden of disease attributable to epilepsy in Sub-Saharan Africa and the critical role of CHWs in increasing access to health services for vulnerable populations, particularly those with stigmatized health conditions, identifying knowledge gaps and clinical strengths is imperative. This study also provided preliminary information on the psychometric properties of a Portuguese QKAP-EPI in Mozambique. Based on the information from this first phase of the study, the CHWs training package for CHWs will be developed, adapted, implemented, and its effect on training of CHWs will be assessed.

## 5. Conclusions

CHW’s knowledge of epilepsy treatment and safety was found to be adequate. On the other hand, knowledge on the causes of epilepsy, attitudes, and certain treatments of seizures was low. Heterogeneity of this knowledge was present, particularly when using different local names for epilepsy (Dzumba and Nzwiti). This study highlights the need to attend to local cultural believes, terms and practices towards epilepsy when designing training programs about epilepsy treatments and prevention.

### Key Points

One hundred and thirty-five CHWs completed a questionnaire that included socio-demographic characteristics and 44-items divided in six subscales pertaining to KAPs towards epilepsy in nine districts of Sofala Province, Mozambique.The internal consistency and factor structure were examined to evaluate the reliability and construct validity of the QKAP-EPI.We evaluated the association between sociodemographic variables and QKAP-EPI subscales using linear regression models.CHWs epilepsy knowledge of medical treatment, and safety/risks were adequate. However, information on the causes of epilepsy, negative attitudes, cultural treatment, and some aspects of epileptic seizures response were low.Heterogeneity in the level of epilepsy knowledge was observed among CHWs in the different Districts of Sofala Province and when using different local names of epilepsy (*Dzumba* and *Nzwiti*).

## Figures and Tables

**Figure 1 ijerph-19-15420-f001:**
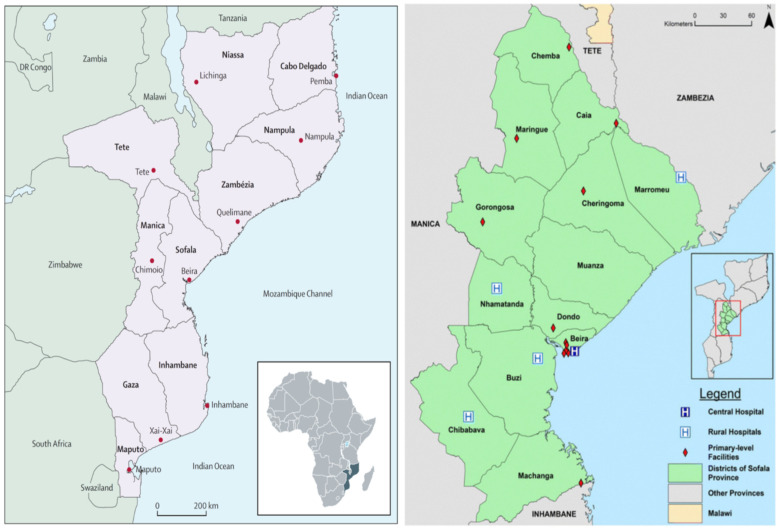
Map of Mozambique and Districts Headquarters Health Facilities in Sofala. Source: adapted from https://www.ncbi.nlm.nih.gov/pmc/articles/PMC4158849 (accessed on 16 May 2022); https://bmcpsychiatry.biomedcentral.com/articles/10.1186/s12888-015-0609-4 (accessed on 16 May 2022). The health facilities above illustrated have at least 1 technician of psychiatry and mental health and at least 1 clinical psychologist in some of them (Caia, Muxungue, Nhamatanda, Dondo, Beira) [24]. Muanza is the closest district to Beira in Sofala, counting on a population of 91.462 inhabitants according to the population projection in 2017 by Provincial Health Directorate of Sofala [25].

**Table 1 ijerph-19-15420-t001:** Sociodemographic description of 135 community health workers in nine districts of Sofala Province, Mozambique.

Variables	Categories	n (%)
Age	20–30	45 (33, 33)
	31–40	41 (30, 37)
	41–50	15 (11, 11)
	>50	33 (24, 44)
	Missing	1 (0, 74)
Sex	Male	94 (69, 63)
	Female	40 (29, 63)
	Missing	1 (0, 74)
Civil Status	Single	47 (34, 81)
	Married	18 (13, 33)
	Common Law Marriage	58 (42, 96)
	Divorced	3 (2, 22)
	Separated	7 (5, 19)
	Widow	2 (1, 48)
Schooling Level Achieved	Primary	44 (32, 59)
	Secondary or High	75 (55, 56)
	Missing	16 (11, 85)
Primary Occupation	Unemployed	63 (46, 67)
	Farmer	31 (22, 96)
	Other (e.g., Students, Teacher, and Trader)	40 (29, 63)
	Missing	1 (0, 74)
Religion	Christian	95 (70, 37)
	Other Religion	36 (26, 67)
	Missing	4 (2, 96)

**Table 2 ijerph-19-15420-t002:** Internal consistency in the subscales of the adapted questionnaire of knowledge, attitudes, practices towards epilepsy and items with low item rest correlation.

Subscale	Ordinal Alpha	Items with Low Item-Rest Correlation ^2^
Causes of epilepsy	0.650	1
Medical treatment ^1^	0.694	13, 14
Cultural treatment	0.797	15, 22
Safety and risks	0.926	—
Negative attitudes	0.904	—
Practices during seizures	0.238	41, 42

Notes: ^1^ Removed item 10 from internal consistency calculation for the medical treatment subscale because not all responses were endorsed and the polychoric correlation matrix would not converge. ^2^ Item descriptions are provided alongside item numbers in Table 3.

**Table 3 ijerph-19-15420-t003:** Proportion of response of each item of the questionnaire of knowledge, attitudes, and practices towards epilepsy of 135 CHWs in Sofala Districts, Mozambique, 2018.

Item of Each Scale	I Don’t Believe n (%)	Believe a Little n (%)	Totally Believe n (%)
**Causes of Epilepsy**			
1.Epilepsy is Inherited	71(52.59)	20 (14.81)	44 (32.59)
2. Head Injury	36 (26.67)	19 (14.07)	80 (59.26)
3. Childbirth Injury	58 (42.96)	26 (19.26)	51 (37.78)
4. Malaria/Meningitis/Fever	50 (37.04)	23 (17.04)	62 (45.93)
5. Brain Injury	28 (20.74)	33 (24.44)	74 (54.81)
**Medical Treatment**			
6. Possible to Treat Epilepsy	12 (8.89)	19 (14.07)	104 (77.04)
7. Antiepileptic Medicines Should Be Taken Continuously to Function Properly	12 (8.89)	18 (13.33)	105 (77.78)
8. Antiepileptic Medicines Are Available at the Health Facilities	7 (5.19)	9 (6.67)	119 (88.15)
9. Njiri Njiri * is Best Treated by a Doctor	18 (13.33)	20 (14.81)	97 (71.85)
10. PWE Should Be Placed in a Safe Place During the Seizures	3 (2.22)	0 (0.0)	132 (97.78)
11. Antiepileptic Medicines Control Seizures	16 (11.85)	22 (16.30)	97 (71.85)
12. Lack of Antiepileptic Medicines May Precipitate Seizures on PWE	17 (12.59)	20 (14.81)	98 (72.59)
13. *Nzwite* * is Best treated By a Doctor	17 (12.59)	16 (11.85)	102 (75.56)
14. Antiepileptic Medicines May Cause Side Effects	42 (31.11)	38 (28.15)	55 (40.74)
**Cultural Treatment**			
15. PWE that Burn Will Never Be Cured	69 (51.11)	21 (15.56)	45 (33.33)
16. *Dzumba ** Can Be Treated Unlike Nzwite	64 (47.41)	32 (23.70)	39 (28.89)
17. *Guru* * is Best Treated by a Nhanga	94 (69.63)	25 (18.52)	16 (11.85)
18. Spreading Water in PWE During Seizures Treats Epilepsy	107 (79.26)	16 (11.85)	12 (8.89)
19. Make Smell Shoes to PWE, During Seizures Treats Epilepsy	107 (79.26)	16 (11.85)	12 (8.89)
20. Fumigation Treats Epilepsy	109 (80.74)	12 (8.89)	14 (10.37)
21. It is Good to Put a Stick in the Mouth of the PWE During Seizures	104 (77.04)	12 (8.89)	19 (14.07)
22. Legs of a PWE Should Be Stretched During Seizures	64 (47.41)	28 (20.74)	43 (31.85)
23. *Dwiti* * is Best Treated by a *Nhanga ***	84 (62.22)	33 (24.44)	18 (13.33)
**Safety and Risks**			
24. PWE Should Not Climb the Trees	26 (19.26)	9 (6.67)	100 (74.04)
25. PWE Should Not Drive	25 (18.52)	15 (11.11)	95 (70.37)
26. PWE Should Avoid Being Near the Fire	15 (11.11)	5 (3.70)	115 (85.19)
27.PWE Should Avoid Staying in Places Near Water	17 (12.59)	11 (8.15)	107 (79.26)
**Negative Attitudes**			
28. PWE Should Not or Can Not Get Married	89 (65.93)	14 (10.37)	32 (23.70)
29. PWE Should Not Go to School	97 (71.85)	8 (5.93)	30 (22.22)
30. PWE Should Not or Can Not Have a Job	100 (74.04)	13 (9.63)	22 (16.3)
31. PWE Should Not or Can Not Have a Normal Life	83 (61.48)	17 (12.59)	35 (25.93)
32. PWE Should Be Isolated	109 (80.74)	8 (5.93)	18 (13.33)
33. PWE Should Be Rejected	109 (80.74)	12 (8.89)	14 (10.37)
34. PWE Should Be Offended	112 (82.96)	9 (6.67)	14 (10.37)
35. PWE are Burden for Society and Family	74 (54.81)	22 (16.30)	39 (28.89)
36. PWE Performs Poorly at School	68 (50.37)	34 (25.19)	33 (24.44)
37. PWE Gives a Lot of Work	70 (51.85)	27 (20.00)	38 (28.15)
38. PWE Are Crazy	84 (62.22)	23 (17.04)	28 (20.74)
**Practices During Seizures**			
39. We Should Put a Stick in the Mouth of the PWE During Seizures	99 (73.33)	14 (10.37)	22 (16.30)
40. We Should Give a Drink to the PWE During Seizures	106 (78.52)	17 (12.59)	12 (8.89)
41. We Should Stay Away from PWE During Seizures	96 (71.11)	14 (10.37)	25 (18.52)
42. We Should Hold PWE During Seizures to Stop Seizures	44 (32.59)	21 (15.85)	70 (51.85)
43. We Must Help PWE During the Seizures Not to Hit the Head on the Ground	14 (10.37)	6 (4.44)	115 (85.19)
44. We Must Place the PWE in the Right Lateral Safety Position to Avoid Aspiration	12 (8.89)	13 (9.63)	110 (81.48)

* Njiri Njiri, Nzwite, Dzumba, Guru, and Dwiti: local terms to designate epilepsy. ** Nhanga: local term to designate traditional healer.

**Table 4 ijerph-19-15420-t004:** Average of subscales and Pearson correlation (r) between subscales of the questionnaire of knowledges, attitudes, and practices towards epilepsy in Sofala, Mozambique.

	Mean (SD)	(1)	(2)	(3)	(4)	(5)	(6)	(7)
(1) Causes of epilepsy	1.10 (0.53)	--						
(2) Medical treatment	1.63 (0.28)	0.170	--					
(3) Cultural treatment	1.48 (0.39)	−0.213	0.007	--				
(4) Safety and risks	1.62 (0.59)	−0.078	0.159	0.108	--			
(5) Negative Attitudes	1.47 (0.48)	−0.118	−0.071	0.392	0.319	--		
(6) Practices of sticking into a mouth, give a drink, stay away	1.43 (0.42)	−0.007	0.132	0.174	0.141	0.151	--	
(7) Practices of holding, helping, placing in safety position	1.60 (0.48)	−0.109	0.054	0.575	0.204	0.254	0.086	--

## Data Availability

The data are available under reasonable request addressed to Vasco Cumbe (vcumbe@gmail.com).

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
