# Peer review of "Community Health Workers’ Knowledge, Attitudes, and Practices towards Epilepsy in Sofala, Central Mozambique"

_ijerph, 2022, doi:10.3390/ijerph192215420_

Round 1

Reviewer 1 Report

The manuscript Community Health Workers' Knowledge, Attitudes, and Practices Towards Epilepsy in Sofala, Central Mozambique presents statistical research based on 135 community health workers (completed a questionnaire).

This study aimed to assess the knowledge, attitudes and practices of community health workers (CHW) towards epilepsy in Mozambique.

CHW's knowledge of epilepsy treatment and safety was found to be adequate. On the other hand, information on the causes of epilepsy, attitudes, and certain treatment of seizures was low.

There is no information in the research

- number of patients with epilepsy / number of patients per one CHW

- propossition for specific educational programs

I believe that the manuscript can be published after minor revisions.

Author Response

##Reviewer 1: The manuscript Community Health Workers' Knowledge, Attitudes, and Practices Towards Epilepsy in Sofala, Central Mozambique presents statistical research based on 135 community health workers (completed a questionnaire). This study aimed to assess the knowledge, attitudes, and practices of community health workers (CHW) towards epilepsy in Mozambique. CHW's knowledge of epilepsy treatment and safety was found to be adequate. On the other hand, information on the causes of epilepsy, attitudes, and certain treatment of seizures was low.

Question 1: there is no information in the research regarding a number of patients with epilepsy/ number of patients per one CHW.

Response: Thank you for making this question. In this phase, we did not explore the number of patients with epilepsy, since the focus was to explore the community health workers' knowledge regarding epilepsy. The next phase of the study is going to be explored the profile of epilepsy attendance in primary health care. There is no data regarding the number of epilepsy patients per CHW since the mentioned CHWs have not been trained yet to attend or refer the epilepsy patients to primary health care.

Question 2: there is no proposition for specific educational programs in the research.  Response: thank you for making this question. The educational programs mentioned in the study would be the guidelines for the training of CHWs (mhGAP: Mental Health Gap Action Plan) in the psychoeducation component about the meaning of epilepsy, its origin, treatment possibilities, and the appropriate places for conventional treatment. We have added a statement in the discussion to reinforce the link between this study and this mhGAP training to better position the study within this context. Additional text: “This study was conducted prior to the implementation of a training in mhGAP to allow trainers to tailor their training activities focused on building capacity for assessment and management of epilepsy within primary care to be appropriate for and responsive to CHW’s current knowledge, attitudes, and practices”.

Reviewer 2 Report

This is an important study because there is little published on epilepsy in Mozambique, especially where community health workers are concerned.

I have a number of comments:

In line 55 the authors state that epilepsy is the most common neurological disease in the world. That is contentious given that migraine for example is a neurological disease.

In line 65 the statement "In low-resource countries epilepsy is often perceived as a mental illness" needs further explanation given that the authors are mostly psychiatrists.

In line 65 a reference on the internet is quoted as saying that treatment costs are less than $5 per year. That would surprise me so further information is needed, in particular the costs of carbamazepine and valproate for example in Mozambique

In Table 3 the first question states that epilepsy is inherited. The correct answer should be "no", but if the statement was "Epilepsy can be inherited", then the answer is "yes".  This needs to be clarified as does the difference between "Brain injury" and "Head injury": many people would regard these as the same unless they were defined precisely.

There is a lot of statistical testing in this paper and I am unconvinced that it adds very much.  Particularly sections 3.2, 3.4, and 3.5 of the resultsIt certainly makes the paper more difficult to read.

One omission is a table of the total and subsection scores in each health worker.  It is possible that a few CHWs were responsible for the low scores in all subsections.

Author Response

##Reviewer 2: This is an important study because there is little published on epilepsy in Mozambique, especially where community health workers are concerned. I have a number of comments:

Question 1: In line 55 the authors state that epilepsy is the most common neurological disease in the world. That is contentious given that migraine for example is a neurological disease.

Response: thank you for this recommendation. We have implemented changes as below:

Original text: “Epilepsy is the most common neurological disease in the world, affecting 50 million people….”

Amended text: “Epilepsy is one of the most common neurological diseases in the world, affecting 50 million people”

Question 2: In line 65 the statement "In low-resource countries, epilepsy is often perceived as a mental illness" needs further explanation given that the authors are mostly psychiatrists.

Response: thank you for this recommendation. In the first half of the 19th century, epilepsy and the insanities were erroneously considered as closely related "neurotic" disorders or whether epilepsy is related to madness. That is why epilepsy is falsely perceived as related to bad spirits or witchcraft. We have implemented changes as below:

Original text: “In low-resource countries, epilepsy is often perceived as a mental illness resulting from spiritual possession or witchcraft”.

Amended text: “In low-resource countries epilepsy is often perceived as a mental illness or related to madness, resulting from spiritual possession or witchcraft,”

Question 3: In line 65 a reference on the internet is quoted as saying that treatment costs are less than $5 per year. That would surprise me so further information is needed, in particular the costs of carbamazepine and valproate for example in Mozambique.

Response: thank you for this recommendation. In the context of Mozambique, considering that more than 90% of hospital care is public and care for chronic diseases is subsidized by the government, and epilepsy treatment is free. We have implemented changes as below:

Original text: “If proper diagnosis and treatment were provided, up to 70% of PWE would live seizure-free. Globally, between 50-75% of PWE are not treated, despite treatment costing less than $5 per year (11),”

Amended text: “If proper diagnosis and treatment were provided, up to 70% of PWE would live seizure-free. Globally, between 50-75% of PWE are not treated, despite treatment costing less than $5 per year (11). In Mozambique, the National Health Service offers care and treatment services free of charge, and more than 90% of the population benefits from these services.”

Question 4: In Table 3 the first question states that epilepsy is inherited. The correct answer should be "no", but if the statement was "Epilepsy can be inherited", then the answer is "yes".  This needs to be clarified as does the difference between "Brain injury" and "Head injury": many people would regard these as the same unless they were defined precisely.

Response: thank you for this recommendation. The first question “epilepsy is inherited” also means “epilepsy is transmitted from father or mother to sons” was translated from the original English questionnaire to Portuguese “Epilepsia é herdada ou transmitida de pais para filhos”.

“Brain injury” is described as trauma in the brain without affecting the scalp and skull, and when translated to Portuguese means “lesão cerebral, while “head Injury” means a trauma affecting the scalp or skull and may sometimes affect the brain.

We have added an extensive discussion of some of the translation, adaptation, and measurement issues in the discussion.

Amended text: First, we used a Portuguese instrument that at some point may not have facilitated the comprehension of the instrument in its entirety by CHWs, because some of them may not fully understand Portuguese and may be more comfortable in their local language. Second, although adapted, this instrument has not been previously validated for the context of Mozambique. Last, the findings may not be generalizable to other areas in Mozambique due to the contextual specificities of the studied area. These limitations may have eventually led to differences in interpretation of the knowledge, attitudes, and practices statements. For example, the ‘correct’ response for some of the knowledge items, such as whether epilepsy is inherited and whether it can be caused by head injury have the potential to vary based on subtle differences in translation and interpretation. For these reasons, we suggest the use of locally adapted and validated instruments in other regions of Mozambique in case of similar study is intended.

Question 5: There is a lot of statistical testing in this paper, and I am unconvinced that it adds very much.  Particularly sections 3.2, 3.4, and 3.5 of the results certainly make the paper more difficult to read.

Response: thank you for this recommendation. Although it is not a study of adaptation and validation of an instrument to assess knowledge, attitudes, and practices towards epilepsy, we believe that it would be important to describe the psychometric properties of the subscales (section 3.2), mean of the subscales (3.5) for giving an overview of the functioning of the instrument used after adaptation. We agree to remove section 3.5 from the manuscript but believe that (given the current limitations around measurement and the use of this scale for the first time in Mozambique) exploring these psychometric properties is important. We have made some revisions throughout the text to improve clarity.

Question 6: One omission is a table of the total and subsection scores in each health worker.  It is possible that a few CHWs were responsible for the low scores in all subsections.

Response: thank you for this recommendation. We think of treating the information in the general context rather than the individualized one. We agree with this possibility raised by the reviewer, but this can be framed as a limitation of the article itself when trying to summarize the manuscript to make it more fluid.

Reviewer 3 Report

The aim of this  study is to assess the Knowledge, attitudes and practices (KAPs) of community health workers towards epilepsy in Mozambique. The abstract is well written with a succinct conclusion. The introduction provided relevant information and pointed out the gap and need of this study. I suhggest to add a statement The aim of this  study is to assess the Knowledge, attitudes and practices (KAPs) of community health workers towards epilepsy in Mozambique." in the end of the last paragraph.

Materials and Method. I suggest to add the sample size calculation, and strategy to improve the response rate.

Results section is long, and can be concise. Tables are not well presented, e.g. the catergories such be mutually exclusive in Table 1. The authors should consult a statisticians or senior researcher for revisions of all tables.

Discussion 

The limitations of the study (questionnaire survey) is not well discussed.

Conclusion

Conclusion is too long. It should be concise and address the aim of the study. 

Author Response

##Reviewer 3: The aim of this study is to assess the Knowledge, attitudes, and practices (KAPs) of community health workers towards epilepsy in Mozambique. The abstract is well written with a succinct conclusion.

Question 1: The introduction provided relevant information and pointed out the gap and need of this study. I suggest adding a statement “The aim of this study is to assess the Knowledge, attitudes and practices (KAPs) of community health workers towards epilepsy in Mozambique." in the end of the last paragraph.

Response: thank you for this recommendation. We have implemented changes as below:

Original text: “This study represents the first phase of assessing the knowledge, attitudes, and practices of CHWs towards epilepsy before evaluating the effect of training CHWs using the WHO Mental Health Gap Action Program (MhGAP) epilepsy package, with the expectation of recommending MISAU for including this adapted package into national CHWs training curricula.”

Amended text: “This study represents the first phase of assessing the knowledge, attitudes, and practices of CHWs towards epilepsy before evaluating the effect of training CHWs using the WHO Mental Health Gap Action Program (MhGAP) epilepsy package, with the expectation of recommending MISAU for including this adapted package into national CHWs training curricula. The aim of this study is to assess the Knowledge, attitudes, and practices (KAPs) of community health workers towards epilepsy in Mozambique”.

Question 2: Materials and Method. I suggest adding the sample size calculation, and strategy to improve the response rate.

Response: Thank you for this suggestion. We have clarified that our sampling strategy was based on a convenience sample. Given the descriptive and exploratory nature of this study, we did not conduct a sample size or power calculation a priori and it is likely that our study would be underpowered for hypothesis testing. The goal of the study is the provide a description of the range in knowledge, attitudes, and practices in our sample and we also note that given the small sample size and the nature of the sampling approach, we cannot ensure that these results are generalizable to CHWs across Mozambique.

  1. a) Original Text: Fifteen CHWs were selected from nine districts (total n=135) in Sofala province (Figure 1). Districts were included if they had no previous implementation of World Health Organization Mental Health Gap Action Program (WHO mhGAP) and had no trained CHWs in the detection and management of epilepsy cases. A Maximum of 20 CHWs are allocated to each district – if districts had more than 15 CHWs we randomly selected 15 from the larger group. The lead CHW in each health facility invited all CHWs at that health facility to participate in the study. Participants were included if they were 18 years or older, belonged to the district headquarters, were fluent in Portuguese, and provided written informed consent.
  2. a) Amended text: “Fifteen CHWs were selected purposefully from nine districts (total n=135) in Sofala province (Figure 1). Districts were included if they had no previous implementation of World Health Organization Mental Health Gap Action Program (WHO mhGAP) and had no trained CHWs in the detection and management of epilepsy cases. In general terms, 15 CHWs are allocated to each district – if districts had more than 15 CHWs we randomly selected 15 from the larger group. The lead CHW in each health facility invited all CHWs at that health facility to participate in the study and there were no refusals reported. Participants were included if they were 18 years or older, belonged to the district headquarters, were fluent in Portuguese, and provided written informed consent.”
  3. b) Original Text: “Given the small sample size and the descriptive, exploratory nature of the study, we did not correct for multiple comparisons “
  4. b) Amended Text: “Given the small sample size and the descriptive, exploratory nature of the study, we did not correct for multiple comparisons, and thus we cannot make generalizations to the larger population of CHWs in Mozambique. “

Question 3: Results section is long and can be concise. Tables are not well presented, e.g. the categories such be mutually exclusive in Table 1. The authors should consult a statisticians or senior researcher for revisions of all tables.

Response: Thank you for this feedback. We have edited the results to simplify their presentation in both the text and the tables and have consulted with a statistician. This includes removing one section from the results (section 3.5) and modifying the way data are presented in some of the tables. We hope this has improved clarity.

Question 4: Discussion (the limitations of the study (questionnaire survey) is not well discussed.

Response: thank you for this recommendation. We have implemented changes as below and elaborated on the key limitations in the study:

Original Text: “First, we used a Portuguese instrument that at some point may not have facilitated the comprehension of the instrument in its entirety by CHWs, because some of them may not fully understand Portuguese and be more comfortable with their local language of the community. Second, although adapted, this instrument has not been previously validated for the context of Mozambique.

Amended text: “First, we used a Portuguese instrument that at some point may not have facilitated the comprehension of the instrument in its entirety by CHWs, because some of them may not fully understand Portuguese and may be more comfortable in their local language. Second, although adapted, this instrument has not been previously validated for the context of Mozambique. Last, the findings may not be generalizable to other areas in Mozambique due to contextual specificities of the studied area. These limitations may have eventually led to differences in interpretation of the knowledge, attitudes, and practices statements. For example, the ‘correct’ response for some of the knowledge items, such as whether epilepsy is inherited and whether it can be caused by head injury have the potential to vary based on subtle differences in translation and interpretation. For these reasons, we suggest use of locally adapted and validated instruments in other regions of Mozambique in case of similar study intended.”

Question 5: Conclusion (conclusion is too long. It should be concise and address the aim of the study). 

Response: thank you for this recommendation. We have implemented changes as below:

Original Text: “In general, the CHWs presented a moderate level of knowledge about epilepsy, with relatively high scores regarding general medical treatment and safety and risks of epilepsy patients. However, CHWs showed gaps in knowledge regarding negative attitudes, culturally specific epilepsy practices, causes of epilepsy, and practices during epileptic seizures. Heterogeneity of this knowledge across different districts of Sofala province was present, particularly when using different local names for epilepsy (Dzumba and Nzwiti). This study highlights the need to attend to local cultural beliefs, local terms, and practices towards epilepsy when designing programs for epilepsy training and prevention and care, and the dissemination of different local terms of epilepsy in different districts. A CHWs training program is required to enhance the level of knowledge, attitudes, and practices about epilepsy and to facilitate the provision of adequate care and treatment of community members with epilepsy. Furthermore, adequate training of CHWs will also improve their bridging role to PHC by strengthening referral and contra-referral of PWE living in communities served by CHWs. “

Amended text: “CHW's knowledge of epilepsy treatment and safety was found to be adequate. On the other hand, knowledge on the causes of epilepsy, attitudes, and certain treatment of seizures was low. Heterogeneity of this knowledge was present, particularly when using different local names for epilepsy (Dzumba and Nzwiti). This study highlights the need to attend to local cultural beliefs, terms, and practices towards epilepsy when designing programs of epilepsy training and prevention”.

Round 2

Reviewer 3 Report

Accept